

# Comparative transcriptome analyses of different *Salvia miltiorrhiza* varieties during the accumulation of tanshinones

Jingwen Zhou[1,2], Rui Liu[1,2], Min Shuai[1,2], Zhu-Yun Yan[1,2] and Xin Chen[1,2]

[1] School of Pharmacy, Chengdu University of Traditional Chinese Medicine, Chengdu, Sichuan, China
[2] Key Laboratory of Characteristic Chinese Medicinal Resources in Southwest, Chengdu, Sichuan, China

Corresponding author
Xin Chen, chenxin@cdutcm.edu.cn

## ABSTRACT

*Salvia miltiorrhiza* (Labiatae) is an important medicinal plant in traditional Chinese medicine. Tanshinones are one of the main active components of *S. miltiorrhiza*. It has been found that the intraspecific variation of *S. miltiorrhiza* is relatively large and the content of tanshinones in its roots of different varieties is also relatively different. To investigate the molecular mechanisms that responsible for the differences among these varieties, the tanshinones content was determined and comparative transcriptomics analysis was carried out during the tanshinones accumulation stage. A total of 52,216 unigenes were obtained from the transcriptome by RNA sequencing among which 23,369 genes were differentially expressed among different varieties, and 2,016 genes including 18 diterpenoid biosynthesis-related genes were differentially expressed during the tanshinones accumulation stage. Functional categorization of the differentially expressed genes (DEGs) among these varieties revealed that the pathway related to photosynthesis, oxidative phosphorylation, secondary metabolite biosynthesis, diterpenoid biosynthesis, terpenoid backbone biosynthesis, sesquiterpenoid and triterpenoid biosynthesis are the most differentially regulated processes in these varieties. The six tanshinone components in these varieties showed different dynamic changes in tanshinone accumulation stage. In addition, combined with the analysis of the dynamic changes, 277 DEGs (including one dehydrogenase, three CYP450 and 24 transcription factors belonging to 12 transcription factor families) related to the accumulation of tanshinones components were obtained. Furthermore, the KEGG pathway enrichment analysis of these 277 DEGs suggested that there might be an interconnection between the primary metabolic processes, signaling processes and the accumulation of tanshinones components. This study expands the vision of intraspecific variation and gene regulation mechanism of secondary metabolite biosynthesis pathways in medicinal plants from the "omics" perspective.

## INTRODUCTION

*Salvia miltiorrhiza* is a plant of the genus *Salvia* in the family Labiatae. Its dried roots and rhizomes are called Danshen in China, and are often used to treat cardiovascular disease (*Dan et al., 2019*) and Alzheimer's disease (*Zhang et al., 2016*). Lipophilic tanshinone is one of the main active components of Danshen (*Ma et al., 2015*), which is derived predominantly from the plastid methylerythritol phosphate (MEP) pathway or partly through the cytoplasmic mevalonate (MVA), and then synthesized through diterpene biosynthesis pathways (*Zhang et al., 2015*), including tanshinone I, tanshinone IIA, tanshinone IIB, cryptotanshinone, and dihydrotanshinone, which are mainly accumulated in the roots and rhizomes of *S. miltiorrhiza* (*Chang et al., 2019*). As far as the general growth trend of *S. miltiorrhiza* is concerned, it can be divided into three stages. The first stage is the vegetative growth period in which the aerial part of *S. miltiorrhiza* grows rapidly. The second stage is the flowering phase, from bolting to flower withering. The third stage is the post-anthesis phase, from the end of flowering to the second vegetative growth period. Previous studies have shown that tanshinones contents began to accumulate in the post-anthesis stage and reached a peak after 60 days in the post-anthesis phase (*Liang et al., 2015*).

*S. miltiorrhiza* is widely distributed and cultivated in China. In the long-term adaptation process to the different local growing environments, it has formed rich genetic diversity. Previous studies about the diversity among varieties of *S. miltiorrhiza* from different habitats was performed by using molecular markers such as inter simple sequence repeats (ISSR), sequence-related amplified polymorphism (SRAP) (*Song et al., 2010*), amplified fragment length polymorphism (AFLP) (*Wang et al., 2007*), expressed sequence tag-derived simple sequence repeat markers (EST-SSR) (*Deng et al., 2009*; *Wang et al., 2011*), CAAT box polymorphism (CBDP) (*Fabriki-Ourang & Karimi, 2019*) and random amplified polymorphism-DNA (RAPD) (*Sunar et al., 2020*). With the rapid development of sequencing technologies, genome sequencing of some *S. miltiorrhiza* varieties has been carried out by *Zhang et al. (2015)*, *Xu et al. (2016)*, and *Song et al. (2020)*. The transcriptome technique is mainly used for the study of the inducing effect (*Gao et al., 2014*; *Ge et al., 2015*), tissue-specific expression (*Yang et al., 2013*), and discovering the key enzyme genes and transcription factors (TFs) (*Song et al., 2017*; *Zhan et al., 2019*) involved in tanshinones and phenolic acids biosynthesis in *S. miltiorrhiza*.
The comparative transcriptomics technique can be used for understanding the differential expression of genes in specific tissues at specific stages, discovering genes related to specific physiological functions, and inferring the physiological functions of unknown genes.

Secondary metabolites of plants are very important for adaptation, and are products of response to environmental stress. Therefore, there is great variation in the type and/or content of secondary metabolites (*Isah, 2019*; *Yang et al., 2018*). Studies have shown that secondary metabolites vary among related species or individuals in the same species, which are related to transcriptome changes and are heritable (*Li et al., 2020*).
The regulation mechanisms of the biosynthetic pathways and transcriptome of plant secondary metabolism have been studied in many closely related species or different

varieties of same species. For example, a comparative transcriptome analysis of two varieties of *Andrographis paniculata* (Burm. f.) Nees. with significant differences in diterpene andrographolide contents revealed that the expression levels of genes related to the biosynthesis of secondary metabolite andrographolide were higher in the genotype with high andrographolide content as compared to that in the low content group (*Patel et al., 2020*). *Matricaria recutita* and *Chamaemelum nobile* are two chamomile species with different chemical compositions, the comparative analysis of their transcriptome revealed that different gene regulation in the synthesis of terpenoid may contribute to the variation in terpenes types and concentrations in these two species. Combining transcriptome analysis with the results of the content and composition of essential oil terpenoids therein can help to understand the mechanisms of differential terpenoid accumulation in different chemotypes of *Cinnamomum camphora* (*Chen et al., 2018a*). The metabolomics study of *S. miltiorrhiza* found that the same variety grown in different places and varieties of *S. miltiorrhiza* grown in the same place had significant variation in their metabolome, and the contents of bioactive components were affected by the growing environment and genotype (*Zhao et al., 2016*). However, there is a lack of comparative analysis based on transcriptome among different varieties of *S. miltiorrhiza* under the same cultivation environment.

An in-depth study on the differences in transcriptional regulation among different *S. miltiorrhiza* varieties can help to elucidate the ecological adaptation mechanism and the regulation mechanism of tanshinones biosynthesis. In this study, *S. miltiorrhiza* from typical production areas were cultivated in the same experimental field for 6 years. Their tanshinones content was determined, then four varieties with significant differences in tanshinones content were selected for subsequent transcriptome analysis. Transcriptional changes during tanshinones accumulation stage were compared while the four varieties were grown under the same environmental conditions, allowing us to investigate the relationship between gene expression regulation and tanshinones accumulation in different varieties of *S. miltiorrhiza*.

## MATERIALS & METHODS

### Plant material and RNA sample collection

The selected four varieties of *S. miltiorrhiza* were collected from Sichuan (SC), Yunnan (YN), Shaanxi (SX), and Henan (HN) (Table 1). All the varieties were asexually propagated through root cutting and cultivated in the experimental field of Chengdu University of Traditional Chinese Medicine for six years. Roots were sampled at 2 days post-anthesis (S1, the early stage of tanshinones accumulation) and 60 days post-anthesis (S2, the late stage of tanshinones accumulation). Each sample had three biological replicates and was frozen in liquid nitrogen as soon as sampled and then stored at −80 °C for subsequent tanshinones contents determination and transcriptome analysis.

### Determination of tanshinones contents in *S. miltiorrhiza* roots

The contents of tanshinones in four varieties of *S. miltiorrhiza* roots at two periods were determined using ultra-performance liquid chromatography. Tanshinones content was
**Table 1 Environmental information for the origin of the varieties.**

| Group | Declaration of origin | Altitude (m) | Latitude (°) | Longitude (°) | Mean annual temperatures (°C) | Mean annual precipitation (mm) |
|-------|----------------------|--------------|--------------|---------------|-------------------------------|--------------------------------|
| SC | Deyang city, Sichuan Province | 784 | 31.01 | 105 | 16.7 | 883 |
| YN | Mile city, Yunnan Province | 1,792 | 24.59 | 103.62 | 17.3 | 987.5 |
| SX | Tongguan city, Shanxi Province | 330 | 34.62 | 110.18 | 12.8 | 625.5 |
| HN | Sanmenxia city, Henan Province | 673 | 32.61 | 111.96 | 12.5 | 848 |

**Note:**
Abbreviations for the four varieties are as follows: SC, Sichuan; SX, Shanxi; YN, Yunnan; and HN, Henan.

determined on an Agilent 1,290 Infinity UPLC system equipped with a diode array detector (DAD, G4212A) using a ZORBAX Eclipse Plus C18 Rapid Resolution HD HPLC column (2.1 mm × 50 mm, 1.8 um, Agilent). The mobile phase consisted of 0.02% (vol/vol) phosphoric acid solution (A) and acetonitrile (B) at a flow rate of 0.4 mL · min$^{-1}$. The UPLC chromatogram was monitored at 270 nm and the column temperature was set at 25 °C. The six bioactive compositions including tanshinone IIB (TNIIB), dihydrotanshinone I (DTNI), tanshinone I (TNI), cryptotanshinone (CTN), tanshinone IIA (TNIIA) and miltrinoe (MTN), were compared with the authorized standard (Chengdu Alpha Biotechnology Co., Ltd., China). Total tanshinone in this study was comprised of TNIIB, DTNI, TN1, CTN, TNIIA, and MTN.

## RNA extraction, library preparation, and sequencing

Total RNA was extracted using Trizol reagent kit (Invitrogen, Carlsbad, CA, USA) according to the manufacturer's protocol. RNA's integrity was confirmed by using the 2,100 Bioanalyzer (Agilent Technologies, Palo Alto, CA, USA) and checked using RNase-free agarose gel electrophoresis. After total RNA was extracted, mRNA was enriched by Oligo(dT) beads. Then the enriched mRNA was fragmented into short fragments using fragmentation buffer and reverse transcribed into cDNA with random primers by using NEBNext Ultra II RNA Library Prep Kit for Illumina (New England Biolabs, USA). Second strand cDNA was synthesized by DNA polymerase I, RNase H, dNTP and buffer. Then the cDNA fragments were purified with QiaQuick PCR purification kit (Qiagen, Germany), end repaired, poly(A) added, and ligated to Illumina sequencing adapters. The ligation products were size selected by agarose gel electrophoresis, PCR amplified, and sequenced using Illumina HiSeq2500 by Gene Denovo Biotechnology Co., Ltd (Guangzhou, China), and 150 bp paired-end reads were generated.

## Data processing, assembly, and annotation

Fastp (*Chen et al., 2018b*) was used to remove sequencing junctions, primers, and low-quality sequences to obtain high-quality clean reads. All downstream analyses were based on high-quality clean data as determined by Q30, meaning that the error rate was less than 0.1%. High-throughput sequencing data were *de novo* assembled using Trinity (*Grabherr et al., 2011*).

The unigenes were annotated using BLASTX program against the NCBI non-redundant protein (Nr) database (http://www.ncbi.nlm.nih.gov), the Swiss-Prot protein database

([http://www.expasy.ch/sprot](http://www.expasy.ch/sprot)), the Kyoto Encyclopedia of Genes and Genomes (KEGG) database ([http://www.genome.jp/kegg](http://www.genome.jp/kegg)). When annotating the unigenes with the BLAST program (E-value threshold was set $<10^{-5}$), the target sequence with the lowest E-value was selected as the best aligning results. To further assign functions to each unigene, Gene Ontology (GO) annotation was performed by the Blast2GO program according to the categories of Cellular component, Molecular Function, and Biological Process ([http://www.geneontology.org/](http://www.geneontology.org/)).

Protein sequences obtained from CDS prediction of unigenes were aligned by Hmmscan (*Thiriet-Rupert et al., 2016*) to search the domain of the plant transcription factors and these unigenes were classified according to plant TFdb3 to predicted transcription factors (*Jin et al., 2014*).

## Gene expression levels calculation and identification of differentially expressed genes

The assembled unigenes were quantified using RSEM (*Li & Dewey, 2011*) software, and the results were normalized to Fragments Per Kilobase of transcript sequence per Million base pairs sequenced (FPKM). For each unigene, the threshold of FPKM >1 was considered as an expressed gene, and the results could be directly used to compare the differences in gene expression.

DESeq2 (*Love, Huber & Anders, 2014*) was used to analysis of significant differential expression in two-group comparisons. An adjusted *P*-value < 0.05 and |log2(FoldChange)| >1 was set as the threshold for a significant differentially expression level. DEGs among four *S. miltiorrhiza* varieties at two stages were identified by comparing each pair of varieties (*i.e.*, HN-SC, HN-SX, HN-YN, SX-SC, SX-YN, and YN-SC). Tanshinone accumulation-related DEGs were identified by comparing the genes expression of each variety during accumulation stages (*i.e.*, SC1-SC2, YN1-YN2, SX1-SX2, and HN1-HN2).

## Statistical analysis

R (version 3.5.1) was used for the basic statistical analysis. To evaluate differences of tanshinones content among the four varieties during the post-anthesis stage, Student's t-test was carried out. The two-way ANOVA method was used to analyze the effect of different stages, varieties and interaction between stages and varieties on tanshinones accumulation. Significant differences were shown to be statistically significant (*P* < 0.01). Spearman's rank correlation coefficient was used to calculate the correlation between gene expression and tanshinone content, and correlation coefficient > 0.70 was set as a significance level.

## Validation of RNA-Seq data by quantitative real-time RT-PCR

The validation was performed with the same RNA samples used for RNA-seq analysis. The three biological replicates of the four varieties of *S. miltiorrhiza* (SC, YN, SX, and YN) at two accumulation stages were analyzed. cDNA synthesis was done by using 400 ng of the total RNA samples with RT Easy^TMII Kit (Foregene CO., LTD., Chengdu, China).

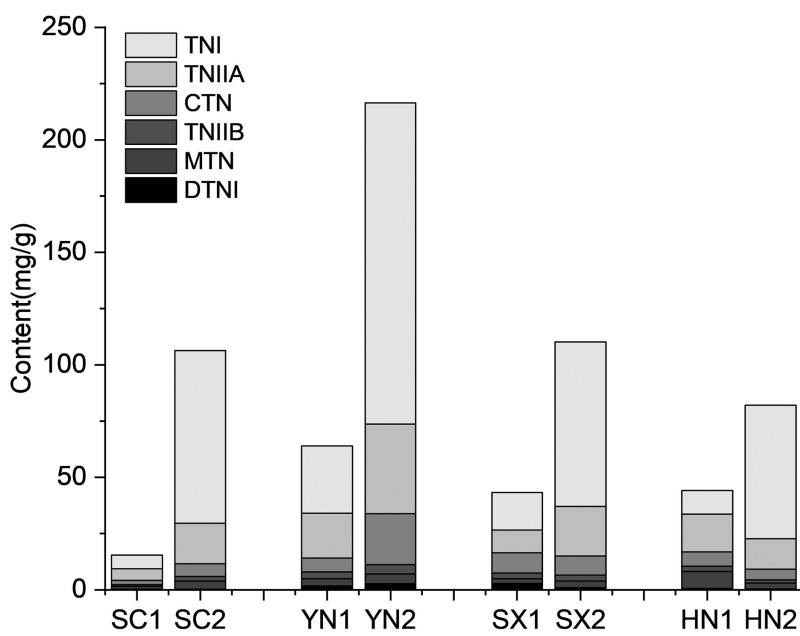

**Figure 1 Tanshinones content in four varieties of *S. miltiorrhiza* at the different stages of tanshinone accumulation.** The X-axis represents four different varieties in two stages, and the Y-axis represents the content of the tanshinone (mg/g). Abbreviations for the four varieties are as follows: SC (Sichuan), SX (Shanxi), YN (Yunnan) and HN (Henan). The number 1 and 2 after the abbreviations for the varieties represent the early stage of tanshinone accumulation (2 days post-anthesis) and the late stage of tanshinone accumulation (60 days post-anthesis), respectively. Abbreviations for the six tanshinones are as follows: TNI (tanshinone I), TNIIA (tanshinone IIA), CTN (cryptotanshinone), TNIIB (tanshinone IIB), MTN (miltrinoe), DTNI (dihydrotanshinone I), and TNI (tanshinone I).

For quantitative PCR, reverse transcribed cDNA products were used as templates. Differentially expressed genes related to tanshinones biosynthesis and candidate transcription factors related to tanshinones accumulation were selected for validation. The primers employed in the qRT-PCR experiments were designed according to the assembled sequences. Primer sequences were reported in Table S1. Actin was used as an endogenous control for the normalization of expression levels of genes (*Jiang et al., 2020*). The reaction was carried out on a CFX Opus Real-Time PCR System using the Real-Time PCR Easy™-SYBR Green I (Foregene CO., LTD., Chengdu, China) with a total reaction volume of 20 μL, 0.4 μM of the primer, and 80 ng of cDNA. Relative gene expression levels were calculated using the $2^{-\Delta\Delta Ct}$ method (*Yuan et al., 2006*). To ensure reproducibility and reliability, three biological replications and three technical replications were implemented for each sample.

## RESULTS

### Determination of tanshinones contents of different *S. miltiorrhiza* varieties during tanshinones accumulation

The tanshinones contents of the four *S. miltiorrhiza* varieties (SC, YN, SX, and HN) were measured at different tanshinones accumulation stages (Fig. 1). The results showed that
the accumulation of tanshinones in the four *S. miltiorrhiza* varieties were different at the two stages under the same environmental conditions (Fig. 1). At S1 stage, the tanshinones contents of each variety were low. The contents of tanshinones in all samples was significantly different between S1 and S2 stage ($P < 0.01$, Table S2), and the content of total tanshinone at S2 was about 6.48 ~ 20.77 times higher than that at S1. By comparing the tanshinones content in different varieties of *S. miltiorrhiza*, it was observed that at S2 the content of CTN, TNIIA, and TNI increased by 7.71, 7.88, and 24.01 times respectively. While, compared with at S1, the content of DTNI, TNIIB and MTN increased only by 3.44, 5.00 and 5.42 times at S2. This phenomenon is consistent with the research previous results of *Yang et al. (2013)*, *Chang et al. (2019)*, *Contreras et al. (2019)*, and *Zhan et al. (2019)* in differential tanshinones contents accumulating in varied degrees. The two-way ANOVA of tanshinone showed that both stage and variety had significant effects ($P < 0.01$) on tanshinones accumulation, and the interaction between them was significant ($P < 0.01$), indicating that the combination of varieties and growth stages had an impact on the tanshinones accumulation (Table S3).

## Sequencing, assembly, and functional annotation of genes in four varieties of *S. miltiorrhiza*

To explore the expression of genes in the four varieties of *S. miltiorrhiza* during tanshinones accumulation, the transcriptome at tanshinones accumulation stages of the four varieties was analyzed by RNA-Seq. A total of 24 cDNA libraries were constructed from root samples with three biological replicates for each stage and each *S. miltiorrhiza* variety after removing the adapter reads. The average GC content of the transcriptome was 41.17%. Each sample yielded more than 5.27 Gb clean data (Table S4). The Q20 and Q30 percentages were more than 92.80% and 75.22%, respectively. The transcriptome sequencing data from the four *S. miltiorrhiza* varieties were filtered and assembled by Trinity resulting in a total of 70,357 unigenes. Overall, the unigenes obtained showed a total assembled bases of 75,359,646, a mean length of 1,071 base pairs (bp) and a mean N50 of 1,911 bp.

These assembled unigenes were searched against the NCBI Nr, SwissProt, KEGG, and KOG databases, and it was found that approximately 56.88% of the unigenes could be annotated in at least one database. The Venn map revealed that 19,454 unigenes were found to be common annotated among four databases (Fig. S1). By sifting out the genes with low expression levels (FPKM < 1), 52,216 genes were obtained (Table S5). KEGG database can link genomic information with functional information, and with KEGG annotation, the functional genes needed for different research can be deeply mined. In this study, a total of 112 genes related to tanshinones biosynthesis were identified, including the terpenoid backbone biosynthesis pathway (Ko00900), diterpenoid biosynthesis pathway (Ko00904) (Table S6).

## Differential gene expression regulation in four *S. miltiorrhiza* varieties

To understand the different regulation of gene expression in different varieties of *S. miltiorrhiza* during tanshinones accumulation stages, we identified the DEGs between

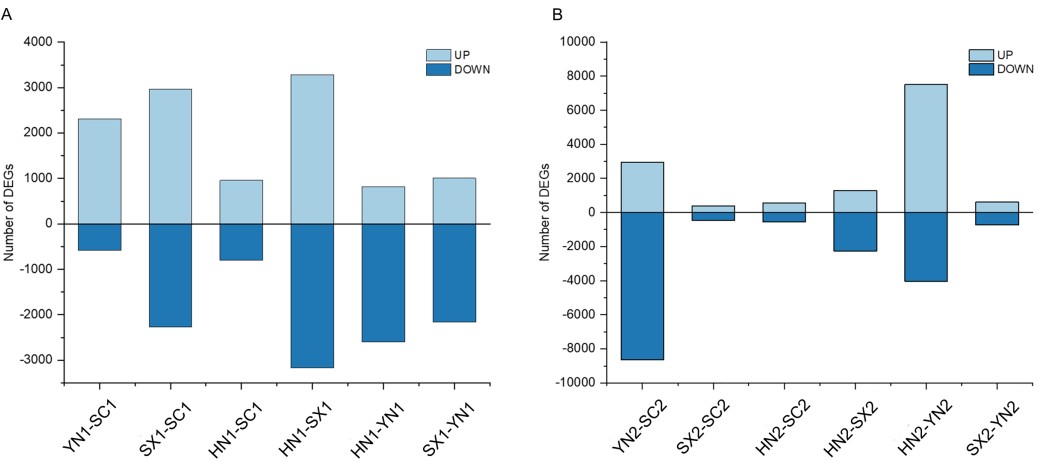

**Figure 2 The summary of gene differential expression among different varieties at the early stage (S1) and late stage (S2) of tanshinones accumulation.** (A) Number of up-regulated and down-regulated DEGs in each pairwise comparison groups at stage S1. (B) Number of up-regulated and down-regulated DEGs in each pairwise comparison groups at stage S2. Abbreviations for the four varieties are as in Fig. 1.       

**Table 2 Number of DEGs in each comparison group at two tanshinones accumulication stages.**

|  | SC | YN | HN | SX |
|---|---|---|---|---|
| SC | – | 11,603 | 1,763 | 853 |
| YN | 2,888 | – | 11,563 | 1,330 |
| HN | 1,119 | 3,417 | – | 3,542 |
| SX | 5,236 | 3,167 | 6,448 | – |

**Note:**
The number of differentially expressed genes among varieties at S1 stage was shown at the bottom left of the table. The number of differentially expressed genes among the varieties at S2 stage was shown in the upper right. Abbreviations for the four varieties are as follows: SC, Sichuan; SX, Shanxi; YN, Yunnan; and HN, Henan.

each pair of varieties (*i.e.*, HN-SC, HN-SX, HN-YN, SX-SC, SX-YN, and YN-SC) at S1 and S2 (Fig. 2). Among the pairwise comparison groups, a total of 11,368 and 17,639 DEGs were detected in S1 and S2 stages respectively (Fig. S3). Only one IREG3 (Unigene 0009899) gene was differentially expressed in all the pairwise comparison groups at S1 (Fig. S3A), while there was no common differentially expressed gene at S2 (Fig. S3B). Additionally, the number of DEGs in each pairwise comparison during tanshinones accumulation revealed that the dynamic regulation of gene expression was varied among the four varieties (Table 2). Except for YN-HN and YN-SC, the number of DEGs at S1 was more than that at S2 in the other four pairwise comparison groups (Table 2).

KEGG enrichment analysis is helpful to further understand the potential function of the genes. In stage S1, DEGs in six pairwise comparison groups were mapped to 100–128 KEGG pathways (Table S7). Of these KEGG pathways, 88 biological processes related to primary metabolism, secondary metabolism, and signal transduction processes were identified in all pairwise comparison groups (HN1-SC1, HN1-SX1, HN1-YN1, SX1-SC1, SX1-YN1 and YN1-SC1) (Table S7). In stage S2, DEGs in six pairwise comparison groups were mapped to 73–136 KEGG pathways, and 51 pathways related to secondary

metabolism and signal transduction were identified in all pairwise comparison groups (Table S8). Of these KEGG pathways, 48 metabolic processes including plant-pathogen interaction, MAPK signaling pathway, galactose metabolism, pyruvate metabolism, amino sugar and nucleotide sugar metabolism, and oxidative phosphorylation were the identified in all pairwise comparison at the two stages (Table S7). At stage S1, DEGs were specifically enriched in some processes of primary metabolism (photosynthesis and citrate cycle) and terpenoid backbone biosynthesis. At stage S2, DEGs were mainly related to metabolic processes (thiamine metabolism and propanoate metabolism) and plant circadian regulation (Table S8).

## DEGs related to the changes of tanshinones content during the accumulation of tanshinones

To comprehensively understand the regulation of *S. miltiorrhiza* gene expression during the accumulation of tanshinones period, the gene expression of the four varieties in two stages (*i.e.*, SC1-SC2, YN1-YN2, SX1-SX2, and HN1-HN2) were compared. A total of 2016 DEGs containing 18 genes involved in the diterpenoid biosynthesis were obtained (Table S9). Among them, 10 DEGs including HSP TF (Unigene0006797), RBG (Unigene0014447), and GAP3 (Unigene0014163) etc. were differentially expressed in all the four pairwise comparison groups (Fig. 3A, Table S9). Compared with stage S1, the expression of these TFs at stage S2 was up-regulated (Figs. 3B, 3C). The KEGG enrichment analysis of 2016 DEGs showed that photosynthesis, secondary metabolites biosynthesis, oxidative phosphorylation, diterpenoid biosynthesis, terpenoid backbone biosynthesis, sesquiterpene and triterpene biosynthesis were the most differentially regulated biological processes during the tanshinones accumulation stages among four varieties of *S. miltiorrhiza* (Fig. 3D).

To identify the tanshinones accumulation-related genes, the correlation between 2016 DEGs with the significant accumulated components (DTNI, TNIIA, and TNI) was analyzed, respectively. Using this method, 277 DEGs correlated with all the three components were obtained, containing one dehydrogenase gene and three CYP450 genes (Table S10, Fig. S4). To understand which categories of these genes were overrepresented, these DEGs were further analyzed by a KEGG enrichment analysis. The KEGG pathway analysis mapped to 63 categories, including photosynthesis, carbon fixation in photosynthetic organisms, galactose metabolism, oxidative phosphorylation, biosynthesis of secondary metabolites, ubiquinone and other terpenoid-quinone biosynthesis, metabolic pathways, MAPK signaling pathway, and plant-pathogen interaction (Table S10). The top 20 KEGG pathways with the highest representation are shown in Fig. 3E. These results suggest that the accumulation of tanshinones is not only related to the process of tanshinones biosynthesis, but also related to other secondary metabolic processes, signal transduction processes, and even primary metabolic processes.

Transcription factors (TFs) are a class of proteins that specifically bind to gene promoters and regulate their expression at different levels. In our transcriptome analysis, a total of 1,420 TFs was identified, which could be classified to 55 families (Table S11). During the tanshinones accumulation stage, a total of 124 differentially expressed TFs were

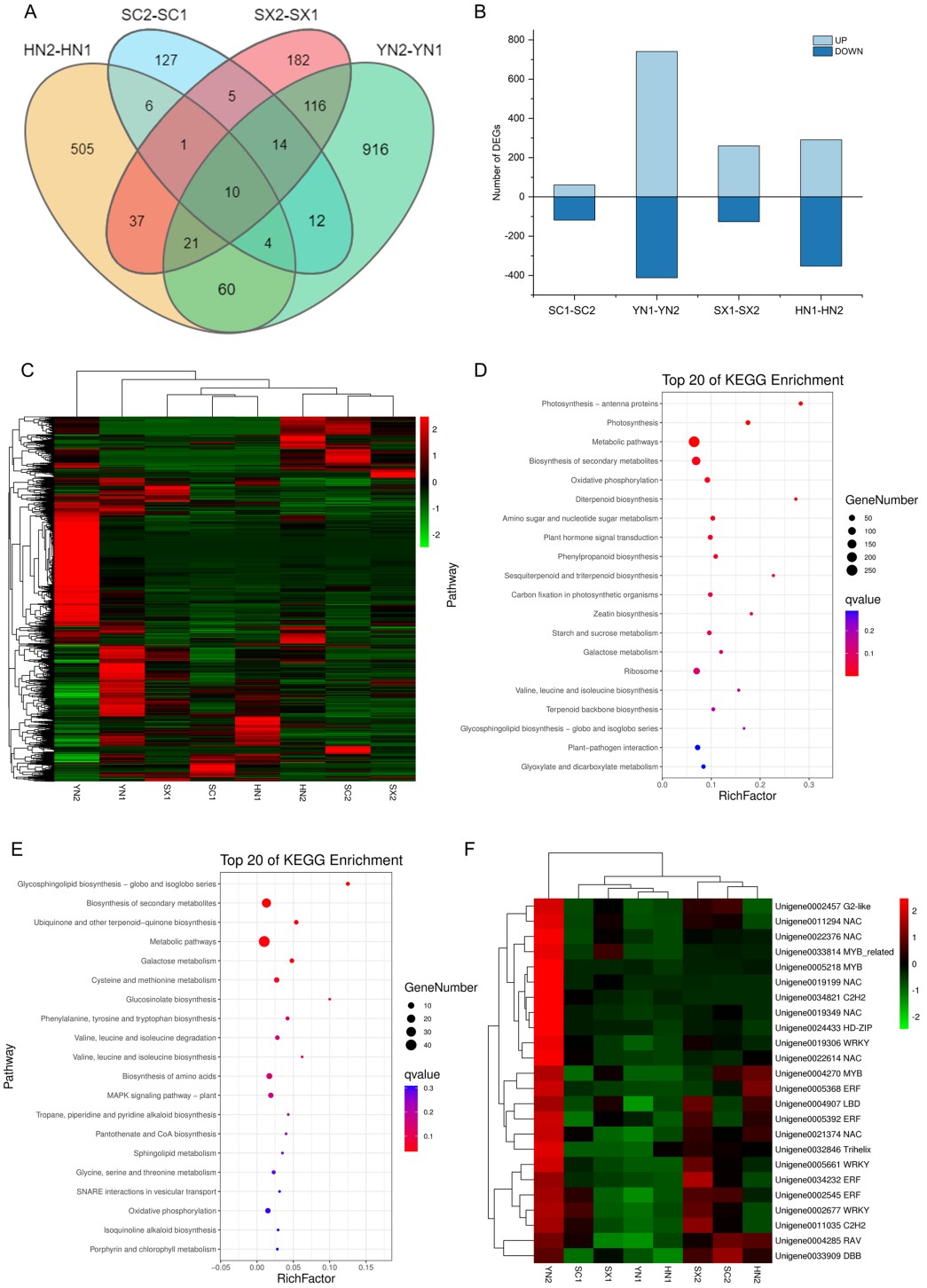

**Figure 3 Differential expression analysis of four varieties of *S. miltiorrhiza* at two stages during tanshinone accumulation.** (A) Total DEGs in the four pairwise comparison group (SC1-SC2, YN1-YN2, SX1-SX2, and HN1-HN2) summarized in a Venn diagram. (B) Overview of up-regulated and down-regulated DEGs of the four varieties of *S. miltiorrhiza* at two stages during tanshinone accumulation. (C) Heatmap of the expression levels of 2016 DEGs. (D) Pathway enrichment analysis of 2016 DEGs. (E) Pathway enrichment analysis of 277 DEGs related to tanshinone accumulation. (F) Heatmap of the expression levels of 24 differentially expressed TFs. Abbreviations for the four varieties are as in Fig. 1.

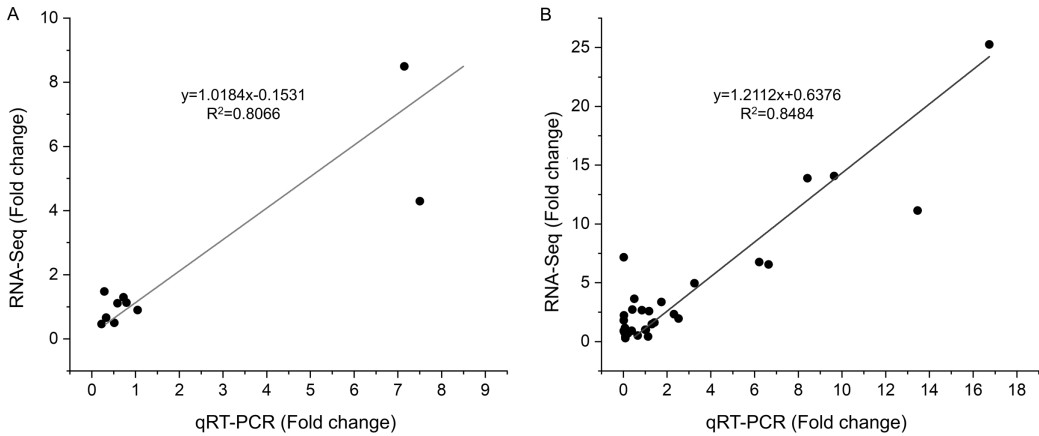

**Figure 4 Correlations of transcript levels of DEGs between RNA-seq and qPCR data at stage S1 and S2.** (A) The correlation of qRT-PCR (X-axis) and RNA-seq (Y-axis) at stage S1. (B) The correlation of qRT-PCR (X-axis) and RNA-seq (Y-axis) at stage S2.

identified in the four varieties (*i.e.*, SC1-SC2; YN1-YN2; SX1-SX2; HN1-HN2) (Table S12). Among the genes associated with tanshinone accumulation, there are 24 TFs, including 6 NAC, 4 AP2/ERF, 3 WRKY, 2 MYB and 1 MYB-related transcription factor (Table S12). These TFs were up-regulated in all varieties of *S. miltiorrhiza* at stage S2, and significantly up-regulated in YN2 with high tanshinones concentrations (Fig. 3F).

## Experimental validation of differential expressed genes by qRT-PCR

To verify the reliability of the RNA-seq results, 10 DEGs at S1 and 10 DEGs at S2 were selected for qRT-PCR analysis (Table S1). These gens included differentially expressed genes related to tanshinones biosynthesis and candidate transcription factors related to tanshinones accumulation (Table S1). The expression levels of these genes were consistent with those determined using RNA-Seq both in stage S1 ($R^2 = 0.8066$) and S2 ($R^2 = 0.8484$) (Figs. 4, S5), indicating that the dataset obtained by RNA-seq was reliable for gene expression analysis.

## DISCUSSION

Besides terpenoid biosynthetic pathway, tanshinones biosynthesis were also associated with primary metabolism, signal transduction and other secondary metabolism processes. A large number of DEGs related to tanshinones accumulation were obtained by the comparative transcriptomics and the tanshinones content dynamic changes analysis. Functional categorization of these DEGs was enriched in terpenoid biosynthesis, photosynthesis, oxidative phosphorylation, and plant hormone signal transduction (Figs. 3D, 3E).

Previous studies have found that the accumulation of secondary metabolites is regulated by cross-talking signaling cascades. Mitogen-activated protein kinase (MAPK) cascades are a prevalent characteristic of eukaryotic cells, involving in plant growth (*Zhang et al., 2018*), development (*Kalapos et al., 2019*), stress response (*Adachi et al., 2015*; *He & Meng, 2020*), and the biosynthesis of secondary metabolites (*De Boer et al., 2011*;

*Devendrakumar, Li & Zhang, 2018*; *Zheng et al., 2019*). In *S. miltiorrhiza*, eighteen MAPKs have been identified, and two SmMAPKs (SmMAPK1, SmMAPK3) might play a role in tanshinones biosynthesis (*Xie et al., 2020*). Among the DEGs associated with tanshinones accumulation obtained in this study, there were six MAPK-related genes, and these genes had higher levels of expression in YN samples with high tanshinones content (Table S10), suggesting their possible role in tanshinones accumulation.

Plants primary metabolism pathways may play an indirect role in secondary metabolic processes. In this study, genes that involved in primary metabolic processes such as photosynthesis and citrate cycle (TCA cycle) were differentially expressed only at the initial stage of tanshinones accumulation (Table S7). Genes related to the regulation of circadian rhythms were specifically differentially expressed at the later stage of tanshinones accumulation (Table S8). In the process of plants responding to environmental changes, primary metabolism processes were affected, and the fixed carbon through photosynthesis becomes allocated to secondary metabolites (*Isah, 2019*). Moreover, plants distributed energy to the secondary metabolic processes through actively inhibits growth in response to environmental changes, which might sometimes involve circadian rhythm response (*Chaves et al., 2002*; *Zhang, Zhao & Zhu, 2020*). These results were consistent with our observations, indicating an interaction between secondary metabolism and primary metabolism. In other species, the reciprocal relationship among secondary metabolism, primary metabolism and signal transduction signaling systems had also been found. For instance, DEGs in two different chemotypes of *Cinnamomum camphora* were mainly associated with carbohydrate metabolism, signal transduction, and terpenoid biosynthesis, and these biological processes-related DEGs resulted in chemotype variation in *C. camphora* (*Chen et al., 2018a*). In *Andrographis paniculata*, the biosynthesis of diterpenoid andrographolide involved the processes such as citrate cycle, carbon fixation, and oxidative phosphorylation in addition to terpenoid biosynthesis (*Patel et al., 2020*). Transcriptional studies of *Daucus carota* with different flavonoids concentrations revealed that the accumulation of flavonoids was related to flavonoid biosynthesis pathway, endogenous signaling and other secondary metabolite biosynthesis pathways (*Meng, Clausen & Rasmussen, 2020*). Additionally, primary metabolism pathways such as galactose metabolism (*Chen et al., 2019*) and starch and sucrose metabolism (*Lloyd & Zakhleniuk, 2004*) also played an indirect role in secondary metabolism.

TFs play a predominant role in regulating the genes expression in various metabolic pathways. The identification of these TFs might be critical for understanding the regulatory mechanisms of tanshinones biosynthesis during tanshinones accumulation stages in different varieties of *S. miltiorrhiza*. The TFs associated with tanshinones biosynthesis mainly belong to AP2/ERF (*Huang et al., 2019*), MYB (*Ding et al., 2017*), bHLH (*Xing et al., 2018*), WRKY (*Li et al., 2015*) and GRAS (*Xing et al., 2018*) family. In the present study, 24 TFs associated with tanshinones accumulation were obtained, including six NAC, four AP2/ERF, three WRKY, two MYB, and one MYB-related TFs. One ERF (Unigene0005368, ERF1B) and three WRKY (Unigene0002677, WRKY24; Unigene0005661, WRKY22; Unigene0019306, WRKY33) TFs related to the MAPK signaling pathway were identified, while one of them, WRKY33, was found to possibly

regulate tanshinones accumulation (*Jiang et al., 2019*), suggesting that the possible regulatory role of the other three MAPK signaling pathway-related TFs as well as other tanshinones accumulation-related (Table S10). In addition, some members of the ERP, bHLH, and MYB transcription factor families have been previously identified to play important regulatory roles in tanshinone biosynthesis and accumulation. Although the NAC family has not been reported in *S. miltiorrhiza* research, studies have identified a regulatory role for NAC transcription factors in terpene biosynthetic pathways (*Jeena et al., 2017*; *Nieuwenhuizen et al., 2015*).

Transcriptome analysis of different varieties of *S. miltiorrhiza* under the same environment is helpful to evaluate the intraspecific variation of *S. miltiorrhiza*. During tanshinones accumulation period, 23,369 (44.75%) of genes were differentially expressed among different varieties of *S. miltiorrhiza*, indicating that there were obvious gene regulation differences among different varieties of *S. miltiorrhiza*. Based on the KEGG pathway enrichment results, a total of 48 co-enriched pathways were observed in each paired comparison group during the tanshinones accumulation period. These common pathways of enrichment involve many physiological processes. For example, "Biosynthesis of amino acids", "carbon metabolism", "galactose metabolism", and "alanine, aspartate and glutamate metabolism" are related to the basic metabolic activities of plants, "plant hormone signal transduction" and "MAPK signal pathway" are related to signal transduction, as well as to the biosynthesis of secondary metabolites such as phenylpropanoid and flavonoid. These functional genes might contribute to intraspecific variations. Among all paired comparison groups, the lowest number of DEGs genes was found between HN and SC varieties, which might be due to a relatively small variation within these two varieties, while correlation analysis between varieties based on all expressed genes also showed a high correlation between HN and SC samples (Figs. 2, S2). Transcriptome sequencing provides a wealth of biological information that has been used to differentiate germplasm in crops such as maize (*Frisch et al., 2010*). Notably, a total of 210 (75.81%) of the content-related genes were also expressed differently in different varieties of *S. miltiorrhiza*, suggesting that the difference of tanshinones content in different varieties of *S. miltiorrhiza* might be related to the variation of tanshinones production. This result was consistent with previous studies in other species. These studies reported that different transcription regulation could distinguish different varieties of the same species, such as, *Peper nigrum* (*Khew et al., 2020*), chamomile (*Tai et al., 2020*), *Cinnamomum burmannii* (*Yang et al., 2020*), *C. camphora* (*Chen et al., 2018a*), *Euphorbia pulcherrima* (*Vilperte et al., 2019*) and *Auricularia auricula-judae* (*Zhao et al., 2019*), and the relative species (*Nieuwenhuizen et al., 2015*; *Tai et al., 2020*; *Wang et al., 2020*).

## CONCLUSIONS

In this study, a comparative transcriptome analysis of four different varieties of *S. miltiorrhiza* during tanshinones accumulation stages was performed. We found that the dynamic change of tanshinones contents in this study are consistent with the phenomenon observed in related studies. By comparing four varieties of *S. miltiorrhiza*,

it was found that there were significant transcriptome differences among varieties in the process of tanshinone accumulation, which may reflect the intraspecific variation of *S. miltiorrhiza*. The functional characterization of genes related to tanshinones accumulation indicated that primary metabolism and signal transduction pathway might play an indirect role in tanshinones accumulation. These studies give us a better understanding of the relationship between primary metabolism, signal transduction and secondary metabolism, and provide a research and application basis for better regulating the overall metabolism of medicinal plants to improve the biosynthesis and accumulation of secondary metabolites. This has also led to a deeper understanding of intraspecific variation in gene regulation in the biosynthetic pathways of plant secondary metabolites.

### Funding

This study was funded by the Natural Science Foundation of China (81973416), and the Chengdu University of Traditional Chinese Medicine characteristic innovative research team project (CXTD2018017). The funders had no role in study design, data collection and analysis, decision to publish, or preparation of the manuscript.

### Grant Disclosures

The following grant information was disclosed by the authors:
Natural Science Foundation of China: 81973416.
Chengdu University of Traditional Chinese Medicine characteristic innovative research team project: CXTD2018017.

### Competing Interests

The authors declare that they have no competing interests.

### Author Contributions

- Jingwen Zhou conceived and designed the experiments, performed the experiments, analyzed the data, prepared figures and/or tables, authored or reviewed drafts of the paper, and approved the final draft.
- Rui Liu performed the experiments, authored or reviewed drafts of the paper, and approved the final draft.
- Min Shuai analyzed the data, authored or reviewed drafts of the paper, and approved the final draft.
- Zhu–Yun Yan conceived and designed the experiments, authored or reviewed drafts of the paper, and approved the final draft.
- Xin Chen conceived and designed the experiments, authored or reviewed drafts of the paper, and approved the final draft.

## DNA Deposition

The following information was supplied regarding the deposition of DNA sequences:

The RNA sequences are available at FigShare: Zhou, Jingwen (2021): Raw data of Sequence. figshare. Dataset. https://doi.org/10.6084/m9.figshare.14298755.v1.

The sequences are available at NCBI, PRJNA712174, and the assembled unigenes are available at NCBI: GJJN00000000.

## Data Availability

Raw data are provided as Supplemental Files.

## Supplemental Information

Supplemental information for this article can be found online at http://dx.doi.org/10.7717/peerj.12300#supplemental-information.

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
