# Peer review of "Comparative transcriptome analyses of different Salvia miltiorrhiza varieties during the accumulation of tanshinones"

_PeerJ, doi:10.7717/peerj.12300_

## Round 0.1 · original submission · Major Revisions

Dear Dr. Chen and co-authors,

As will see, our reviewers found that your work was important and interesting. However, they provided several valuable comments and suggestions to improve your manuscript.

Reviewer 1 had concerns about the quality of some figures and the English writing style. (S)he also suggests that the authors explain a clearer methodology and the background information, including the general life stages of the species and its secondary metabolite accumulation. Reviewer 2 provided several important suggestions, including improvement of Figures 1 and 4. This reviewer also suggests that you should clarify the relationship between the related gene expression and the thashinone levels.

I agree with their comments and suggestions. I think that adding new analysis and/or an in-depth data interpretation makes your paper stronger. I would like to ask you to address or to respond with reasons not to follow the suggestion made by these reviewers.


Best regards,
Atsushi Fukushima

Reviewer 1 ·

Basic reporting

The basic reporting of this manuscript has to be improved intensively:
1. The English or writing of the manuscript has to be improved. For example:
• Line 63: “…. in the family Labiatae, its dry roots and rhizomes…” should be separated into two sentences
• Lines 76-77: change “…Zhang (Zhang et al. 2005), Xu (Xu et al. 2016)… carried out…” to “Zhang et al. (2005), Xu et al. (2016)…carried out…”
• Line 128: “….contents were originated? collected? from Sichuan…”
• Line 293: “Totally 1420 transcription…”
• Line 294: “..675 differentially expression TFs..”
• Other grammatical changes: suggest to be reviewed by a native English speaker
2. The quality and resolution of some of the figures have to be improved (e.g. Fig. 4). However, I am not sure if this is due to the reduced resolution in the reviewed copy of the manuscript or due to the original image submitted.
3. The literature review of the study of secondary metabolite biosynthetic mechanism is too long (i.e. lines 93-106) and covering different types of plants. If possible, it should be focusing on research related to the Salvia genus or its family/order.
4. The introduction is not being well written to give the readers sufficient background information about this research and methodology. For example:
• As the research was done on S. miltiorrhiza sampled at early and late stages of tanshinones accumulation, the general life stages of the plant and its secondary metabolite accumulation (i.e. tanshinones) should be briefly reviewed. This should be linked to the rationale of selecting the early and late stages for this study.
• This research was conducted on S. miltiorrhiza with different varieties, ecological differences and genetic diversity. However, the background information S. miltiorrhiza variety, genetics and ecology was not being described, with only general information of “The strong ecological …. formed rich genetic diversity.” from lines 67-70. This should be linked to the rationale of selecting the specific 4 varieties for this study.
• The authors should also elaborate further on tanshinones, such as its components, chemistry, properties or biosynthesis process.
5. Fig. 2A: The values for S1 (5730) and S2 (12001) are the sums for all 4 varieties? Please revise the figure caption to make it clearer for readers
6. Fig 4: black bars and red lines?

Experimental design

The research question is well-defined but a clearer methodology must be presented:
1. The rationale of selecting the 4 varieties and 2 different stages, as mentioned in point 4 in Basic Reporting, should be enhanced.
2. Some additional information is required for methodology. For example,
• Line 130: how long is the cultivation being done?
• Line 131: which part of the plants was sampled? Roots?
• Line 154: provide information on reverse transcription, enrichment method and library preparation, i.e. kits and manufacturers
• Line 156: provide information on RNA sequencing e.g. paired-end sequencing? Read length? Throughout?
• Line 162: what is the QC stringency used for Q30? Additionally, the use of Q30 (e.g. >90%) in clean reads should be mentioned prior to the Trinity assembly.
• Line 167: how to define “best aligned”?
• Line 171: what is the method for transcription factor prediction?
• Line 175: elaborate “Gene expression was calculated.”. How is it being done, e.g. mapping method and software, criteria, against de novo Trinity transcriptome?
• Line 187: “thresholds above 0.70” = P<0.30?
• Line 190: change “The experiments were performed…” to “The validation was performed…”
• Line 191: why only samples for stage S2 were included in the validation?
• Line 192: 400 ng of total RNA or mRNA?
• Line 193: remove “First-strand cDNA…Real-time PCR”
• Line 194: what is the starting material amount for reverse transcripted template?
• Line 196: any reference(s) of using actin as a housekeeping gene in S. miltiorrhiza as a number of S. miltiorrhiza transcriptome studies have been reported. This is to support the “suitability” of using actin (as traditional housekeeping might have variable expression as well) and a single housekeeping gene (as most studies use 2-3 control genes) for normalisation
3. The authors have mentioned that the whole genomes of S. miltiorrhiza are available (3 different studies). Therefore, the authors should provide the rationale for using de novo transcriptome assembly instead of genome-based reference transcriptome mapping.
4. The methodology and comparison group for RNA-Seq results should be indicated in Materials and Methods, e.g. read mapping and comparison group for 4 different variety versus 2 stages, how it is being done?
5. The authors should elaborate on selection criteria for DEGs for real-time PCR validation and what are the criteria for primer design. How about reference or housekeeping genes?

Validity of the findings

Data is sufficiently provided but the presentation and result description has to be improved:
1. Line 206: cite Fig 1 in the first sentence
2. Fig. 1: the differences between stage 1 and stage 2 should be tested with a suitable statistical test such as t-test and defined as significant at P<0,05 or P<0.01 level. Additional statistical test such as two-way ANOVA may provide statistical information into effect of stages, variety and interaction between stages and variety on tanshinones accumulation
3. Line 210: the phrase “…increased to some extent” is not professional and scientific. In general, it should be described as “increased by XX-XX fold for all XXX except for XXXXX”, or “increased by XX-XX % for all XXX except for XXXXX”.
4. Lines 211-213: the description of “did not change”, “increased greatly” and “increased significantly” is weak without support from the statistical test
5. Line 214: are DTN, TNIIB and MTN being identified (or being proposed?) as the intermediates based on cited studies? The authors have to be careful about the use of “maybe” if it is already a fact or proposed fact by others. Again, this link to the needs for a brief description of tanshinones biosynthesis in the Introduction section
6. There is no comparison on tanshinones profiling among the four varieties, which should be the key focus of this study. How different are the tanshinones content and components among the four varieties? This is not being answered and well presented in Figure 1, although the data is already there. This is important as the comparison is essential to be correlated with the RNA-Seq data for interpretation.
7. Line 225: change “analysed by Trinity” to “assembled by Trinity”
8. Line 228: 19,4454 unigenes?
9. Line 240: do not start a sentence with a number
10. Line 271: any examples of the 10 DEGs common to all 4 comparison groups? At least some interesting candidates to be listed in the result description?
11. Lines 272-273: “the expression of many DEGs …. than at stage 1” indicates an “up-regulation”? Please make it clearer by improving the sentence
12. Lines 294-298: how many TFs being differentially expressed? 675 or 124?
13. Line 308: what is the degree of consistency (quantitative) between RNA-Seq and real-time PCR?
14. Lines 419-422: the statement is too sceptical and not conclusive if it is just based on the current results. It should be indicated as a speculation

Additional comments

The present manuscript is interesting as it targets to investigate the transcriptional and phenotypic differences among 4 different Salvia miltiorrhiza varieties with medicinal properties at 2 different metabolite accumulation timepoint. The current manuscript is a very beginning draft where various improvement has to be made before it can be evaluated further and reconsidered for publication.

Reviewer 2 ·

Basic reporting

1. Figure 1
It would be more clear understanding if showing the concentrations of thashinones at two stages in 4 varieties of S. militiorrhiza in one histogram.
2. The text within Figure 4 is not clear enough.
3. It’s necessary to show the relationship between the expression levels of related genes and the concentrations of thashinones. It would be helpful for understanding the contribution of different transcriptional factors and functional genes in tanshinone accumulation.
4. Line 98
“Lycium Barbaru and L. Ruthenicum” should be “Lycium barbaru and L. ruthenicum”.

Experimental design

no comment

Validity of the findings

no comment

Additional comments

The authors showed comparative transcriptome analysis of four different varieties of S. miltiorrhizaduring tanshinones accumulation. That’s of interest but the manuscript could be improved.

---

## Round 0.2 · Minor Revisions

Dear Dr. Chen and co-authors,

Would you please have a look at the comments and improve your manuscript accordingly?

Best regards,
Atsushi Fukushima

Reviewer 1 ·

Basic reporting

See General Comments to Author

Experimental design

See General Comments to Author

Validity of the findings

See General Comments to Author

Additional comments

Overall, the manuscript has been improved significantly and is ready to be considered for acceptance with minor revision based on the following comments:

Major comments:
1. Provide some sequencing and assembly statistics of the RNA-Seq data, such as how many throughput (Gb of data) per sample, total length (bp) of the assembly, GC content and N50 of the assembly, average unigene length etc. It could be summarised in one or two sentences in the Results section.
2. The grammar has to double check another round as some minor mistakes still exist although the language for the overall manuscript has been improved.
3. The author should add one or two sentences in Conclusion on the significance of the present research, i.e. how the results are useful for future work or applications

Minor comments:
1. Line 40: formatting error, "among these varieties, The tanshinones content"
2. Line 75: Perhaps can make " S. miltiorrhiza is widely distributed and cultivated in China...." into a new paragraph
3. Line 157: Indicate the cDNA kit used (Brand, Country)
4. Line 169: "were co-assembled." into ?? contig? unigene?
5. Lines 191-199: these sentences can be in the same paragraph
6. Line 219: are designed > were designed
7. Line 216: are used > were used
8. Line 223: "0.8 μL of the primer, and 1 μL of cDNA", do not use μL as it is meaningless, use μM/mM or primers and μg or ng for cDNA template
9. Line 239: DTN or DTNI?
10. Lines 237-240: the increment mentioned here is referring to the comparison between S1 and S2?
11. Line 243: "had significant effects", indicate the P value, P<0.05? P<0.01?
12. Line 282: " 73 ~ 136 KEGG", ~ or -?
13. Line 336: "accumulation(Table S1)" - spacing and full stop
14. Figure 4: what is mean by y-axis and x-axis? fold change? log2 fold change? Most of the "DEGs" are below the value 1 which is not optimal, the validation should cover a wide range of dynamic value
15. Line 406: change to 23,369 (44.75%). Same for Lines 421-422
16. Line 433: change to "we found that"
17. Figure 2: " two stages (S1 and S2)", suggest to change to early stage (S1) and late stage (S2) of tanshinones accumulation
18. Line 161: the information of 150 bp read is not mentioned
19. Line 172: "cDNA was synthesized"
20. Line 165: "comprised of"
21. Line 183: "...data were de novo..."
22. Line 182: high quality or high throughput? choose only 1. There is typo for throughput
23. Line 265: degrees ...... previous research results
24: Line 282: one database

Reviewer 2 ·

Basic reporting

See general comments

Experimental design

Well done.

Validity of the findings

Well done.

Additional comments

1. The author wrote in the manuscript: "(SmMAPK1, SmMAPK3) might play a role in tanshinones biosynthesis (Xie et al. 355 2020). Among the DEGs associated with tanshinones accumulation obtained in this study, there were six MAPK-related genes s, and these genes had higher levels of expression in YN samples 357 with high tanshinones content (Table S10)". It is stated that there are 6 SmMAPKs highly expressed in YN samples. According to Xie et al. (2020) study, we know that SmMAPK1 and SmMAPK3 are associated with tanshinone synthesis. So are there SmMAPK1 and SmMAPK3 among the 6 MAPKs stated by the authors and which one is it respectively.

2. The authors quantified the expression of the SmMYC. We know that there are two SmMYC2 in Salvia, SmMYC2a and SmMYC2b, which act as positive regulators of the synthesis of tanshinones and phenolic acids, so which one is the authors' SmMYC2? This needs to be clearly stated.

3. The authors quantified the expression of SmMYB36. Is this gene SmMYB36 as reported by Ding et al (2017)? According to the report, we know that this gene is a positive regulator of tanshinone synthesis and a negative factor for phenolic acids. If it is SmMYB36, the authors need to explicitly specify. However, we checked the primers (Forward primer: AAGAGCTGCAGGTTGAGATG; Reverse primer: GTAACTGAGATGCTATGACCGAC) for the quantitative PCR of the SmMYB36 gene, and this primer did not match the sequence reported by Ding et al (2017). Please explain why this primer does not match the sequence reported by Ding et al (2017).
>SmMYB36 Ding et al. 2017 KF059390
ATGGCGAGTGATGCATCTCTGAACAAAGGGGCGTGGAGTGTGGATGAAGATTCGACTCTGGCCCAATACGTCGCCCTTCACGGCCCCAAGAGGTGGAAATCTGTGGCCATCAAATCAGGCCTCAACAGGTGCGGTAAGAGTTGCAGGTTGAGATGGCTCAACTATCTTAGTCCTGATATCAAAAGAGGCAACTTCTCCGATGCTGAAGAGGACTTGATTCTTAGGTTACATAGGCTCCTAGGAAATAAGTGGTCGTTGATCGCCAAGAGAATCCCTGGGCGGACTGACAACGAGATCAAGAACTACTGGAATGCTCACTTGAGGAAGAAAGCCATGTTGATGGACAAGTTACCCGCAATCTCAACCGCAGCAATGAAGCAAAACATTTGGAATGAAACCGGAGGCGACGATGAATCCTTGGACGTCTCTGCCTCTGGGTTGGATTGGGTTAAAAGTTTTCTTGAACTCGACGAGGATGAATGA

---

## Round 0.3 · Minor Revisions

Dear authors,

Reviewer 1 still had some very minor issues. Would you please improve your manuscript accordingly

In addition, you should check all the citation information and the main text thoroughly.

Best regards,
Atsushi Fukushima

Reviewer 1 ·

Basic reporting

See Additional Comments.

Experimental design

See Additional Comments.

Validity of the findings

See Additional Comments.

Additional comments

Minor comments:
1. The previous comment about providing the cDNA synthesis kit name has not been addressed. This should be different from the PCR purification kit where the author highlighted in their rebuttal.
2. Line 251: Are you sure such a high amount of cDNA (80 micrograms) was used in a single 20u PCR reaction?
3. Line 43: ....that are responsible for the differences...
4. Line 67: dried roots?
5. Line 125: metabolome
6. Line 259: ...were low.
7. Line 409: responding

Reviewer 2 ·

Basic reporting

The manuscript could be accepted.

Experimental design

The manuscript could be accepted.

Validity of the findings

The manuscript could be accepted.

Additional comments

The manuscript could be accepted.

---

## Round 0.4 · Minor Revisions

Dear authors,


Thank you for revising. However, our Section Editor has commented as follows.

"Some minor issues.

+++ The assembled unigenes also need to be deposited at NCBI

+++ What is the planting design? Each variety in a separate block, or was it randomized? if each variety is in a separate block, then the authors need to acknolwedge that and the possibility that differences are due to block differences instead of genetic differences.

+++ line 172 which program was used for co-assembly across varieties?

+++ line 195. Raw reads should be used for DEseq, not FPKM. If FPKM were used the analysis needs to be redone. Otherwise specify the input for DEseq."


Would you please revise the manuscript? Regarding the unigenes you obtained, the authors might want to deposit them into TSA (https://www.ncbi.nlm.nih.gov/genbank/tsa/) before publication.


Best regards

---

## Round 0.5 · accepted · Accept

Dear authors,

Thank you for your reply.

Best regards